

# Effects of adding dual-task or sport-specific task constrains to jump-landing tests on biomechanical parameters related to injury risk factors in team sports: a systematic review

Sara González-Millán[1], Víctor Illera-Domínguez[1], Víctor Toro-Román[1], Bruno Fernández-Valdés[1], Mónica Morral-Yepes[1,2], Lluís Albesa-Albiol[1], Carla Pérez-Chirinos Buxadé[1] and Toni Caparrós[2,3]

[1] Department of Health Sciences, Research Group in Technology Applied to High Performance and Health, TecnoCampus, Universitat Pompeu Fabra, Mataró, Barcelona, Spain
[2] National Institute of Physical Education of Catalonia (INEFC), University of Barcelona, Barcelona, Barcelona, Spain
[3] Sport Research Institute, Universitat Autònoma de Barcelona, Bellaterra, Barcelona, Spain

Corresponding author
Víctor Illera-Domínguez,
villera@tecnocampus.cat

## ABSTRACT

**Background**. Jumping and landing tests are frequently used as a tool to assess muscle function. However, they are performed in a controlled and predictable environment. The physical tests commonly used as part of the criteria for return to sport after injury are often performed with little or no cognitive load and low coordinative demand compared to game-specific actions. The aim of this systematic review was to examine the influence of performing a dual task (DT) or sport-specific task constrains during jump-landing tests on biomechanical variables related to lower limb injury risk in team sports.

**Methods**. This systematic review followed the specific methodological guidelines of the Preferred Reporting Items for Systematic Reviews and Meta-Analyses (PRISMA). The search was conducted in the databases Medline (PubMed), Web of Science, Cochrane Plus, and SportDiscus for studies published from 2013 until June 30, 2023. To be eligible, studies had to include: (1) kinematic and/or kinetic assessment of injury risk factors in the lower extremity; (2) a comparison between a simple jump or landing test and a DT jump or landing test which included cognitive information. The risk of bias in the selected articles was analyzed using the recommendations of the Cochrane Collaboration.

**Results**. Of the 656 records identified, 13 met the established criteria. Additionally, two more articles were manually included after screening references from the included articles and previous related systematic reviews. Regarding the Risk of bias assessment, 12 studies did not surpass a score of 3 points (out of a total of 7). Only three studies exceeded a score of 3 points, with one article achieving a total score of 6. From the included studies, comparative conditions included actions influenced by the inclusion of a sports ball ($n = 6$), performing tasks in virtual environments or with virtual feedback ($n = 2$), participation in cognitive tasks ($n = 6$), and tasks involving dual processes ($n = 7$). The execution of decision-making (DM) during the jump-landing action resulted in biomechanical changes such as lower peak angles of hip flexion

and knee flexion, along with increased vertical ground reaction force, knee abduction, and tibial internal rotation. Regarding limitations, discrepancies arise in defining what constitutes DT. As a result, it is possible that not all studies included in this review fit all conceptual definitions of DT. The inclusion of DT or constraints in jump-landing tests significantly alters biomechanical variables related to lower extremity injury risk in team sports. In future research, it would be beneficial to incorporate tasks into jumping tests that simulate the specific cognitive demands of team sports. This systematic review was registered in PROSPERO (registration number: CRD42023462102) and this research received no external funding.

## INTRODUCTION

Performance in team sports is a multifactorial phenomenon that depends on physical, coordinative, and cognitive factors, as athletes must adapt their movements and actions to various unpredictable and changing game situations, generated, among others, by opponents, teammates, or the ball (*Abernethy, Thomas & Thomas, 1993*). This involves performing motor actions such as jumps, landings, and changes of direction while simultaneously controlling and responding to multiple stimuli, therefore processing large amounts of information within a limited time frame (*Hughes & Dai, 2023*). Furthermore, on certain occasions, multiple motor actions must be executed concurrently (*Moreira et al., 2021*; *Evans & Stanovich, 2013*). Consequently, the ability to allocate attention among the various perceptual demands arising from the environment and the action itself is crucial for successful decision-making (DM).

It is widely accepted that the information processing capacity of the central nervous system is limited, thereby restricting the ability to allocate attention to different inputs and the capacity to prepare for and perform multiple tasks simultaneously (*Marois & Ivanoff, 2005*). The term 'Dual-Task' (DT) is the prevailing expression in scientific literature for denoting the concurrent execution of two or more tasks (*Moreira et al., 2021*; *McIsaac, Lamberg & Muratori, 2015*). This phenomenon has been studied from various research fields using different paradigms, such as executive control, allocation of resources, task prioritization, task switching, and task type (*McIsaac, Lamberg & Muratori, 2015*; *Mas, Naranjo & Mollá, 2023*). Despite the diverse range of theoretical models, there is consensus that during DT, the neurophysiological process of allocating attention across different activities results in increased challenge, making the management and performance of one or both tasks more difficult (*McIsaac, Lamberg & Muratori, 2015*; *Akin et al., 2021*). The effect on task performance during DT can be quantified using various methods, often expressed as cost or the effect one task has on the other: DT Effect $= [(\text{dual} - \text{single})/\text{single} \cdot \pm 100]$ (*Moreira et al., 2021*; *McIsaac, Lamberg & Muratori, 2015*). Various models of DT have traditionally been studied, including Motor-Motor, in which two coordinative or motor tasks are combined, and Motor-Cognitive, in which a motor task is combined with

cognitive processing (*McIsaac, Lamberg & Muratori, 2015*; *Mas, Naranjo & Mollá, 2023*; *Akin et al., 2021*; *O'Shea, Morris & Iansek, 2002*). Furthermore, task constraints such DM which implies reacting to a stimulus or modulating the response based on the stimulus, also influence task performance and are dependent on factors such as time available for reaction, the number and type of stimuli, and the number and complexity of response options available (*Hughes & Dai, 2023*).

The assessment of physical performance is crucial and an integral part of both the sports and rehabilitation domains. Jump-landing tests are frequently used as a tool to assess lower limb muscle function due to their ease of implementation, high reliability, validity, and sensitivity (*Meylan et al., 2009*; *García-Ramos et al., 2023*). Jumping is a motor skill that demands significant coordination of both upper and lower extremities, making it fundamental in a wide variety of sports (*Zhou et al., 2020*; *Rodríguez-Rosell et al., 2017*). There is a wide range of jump tests, which are commonly classified as horizontal jumps (HJ) or vertical jumps (VJ) based on the force application vector. Some of the most frequently assessed HJ tests include the single-leg hop test (SLHT), double-leg hop test (DLHT), triple single-leg hop test (TSLHT), triple double-leg hop test (TDLHT), and the crossover hop test (CHT) (*Meylan et al., 2009*). On the other hand, among the commonly used VJ tests are the squat jump (SJ), countermovement jump (CMJ), Abalakov jump (AJ), and the drop jump (DJ) (*Markovic et al., 2004*; *Sattler et al., 2012*; *Karatrantou et al., 2019*). Jump tests, both VJ and HJ, are also regularly employed for the assessment of biomechanical variables associated with higher risk lower-limb injury patterns (*e.g.*, increased knee valgus, increased internal rotation of the hip, and increased external rotation of the tibia) (*Read et al., 2016*; *Redler et al., 2016*; *Munro & Herrington, 2011*; *Reid et al., 2007*). Additionally, they are frequently used to evaluate athletes after lower limb injuries as part of the criteria for returning to sports practice (*Dutaillis et al., 2023*). For instance, a shorter distance in HJ tests is associated with lower isokinetic knee extension torque (*Järvelä et al., 2002*) and a higher risk of re-injury (*Müller et al., 2015*). In triple hop tests, prolonged amortization indicates reduced capacity to absorb and regenerate ground reaction forces upon landing (*Lloyd et al., 2020*). Moreover, several studies have confirmed that evaluating landing during VJ provides valid and reliable indicators of anterior cruciate ligament (ACL) injury risk factors, such as knee valgus and excessive knee abduction torques (*Hewett et al., 2005*; *Leppänen et al., 2017*; *Padua et al., 2009*; *Hewett & Myer, 2011*). Neuromuscular control deficits can lead to excessive strain on passive ligamentous structures, exceeding their load-bearing capacity and increasing the risk of injury or compromising structural integrity (*Read et al., 2016*).

While bilateral jump-landing tests in a controlled and stable environment are commonly used to evaluate team athletes, the external validity of these tests in measuring lower limb muscle functionality in situations of greater specificity has been questioned (*Rodríguez-Rosell et al., 2017*; *Currell & Jeukendrup, 2008*). It is plausible that in a highly controlled environment devoid of uncertainty, such tests may not reveal any bilateral asymmetries or patterns indicative of injury risk. Yet, these factors might become evident when the tests are conducted under conditions with elevated cognitive or coordinative demands (*Dutaillis et al., 2023*; *Millikan et al., 2019*). The reduction of conscious attention directed towards
the control of the primary task affects the integration of essential visual, auditory, and somatosensory stimuli necessary for neuromuscular control, coordination, and stability (*Masters & Maxwell, 2008*; *Lohse, Sherwood & Healy, 2014*; *Lohse et al., 2014*; *Schnittjer et al., 2021*). Thus, DM and divided attention could significantly influence some biomechanical variables related to a higher risk of ACL injury, such as reduced knee flexion at initial contact, increased knee valgus angles, greater knee extension, and valgus torques (*Hughes & Dai, 2023*).

The impact of cognitive factors on the occurrence of injuries related to team sports is increasingly being investigated (*Shultz et al., 2019*). Traditionally, more emphasis has been placed on anatomical, biomechanical, and hormonal factors in lower-limb injury prevention programs (*Zamankhanpour et al., 2023*). However, these programs do not incorporate key aspects of injury risk, such as reaction time, processing speed, and visual and verbal memory, which are highly relevant in team sports (*Swanik et al., 2007*; *Wilkerson, 2012*). Additionally, the physical tests commonly used as part of the criteria for returning to sports practice after an injury are often conducted with little to no cognitive load and low coordinative demand compared to the specific game actions (*Millikan et al., 2019*). Hence, new proposals for return-to-sport criteria suggest incorporating DT assessments and coordinative and cognitive constraints into the tasks to simulate the unpredictability and divided attention encountered in competition (*Wilk et al., 2023*; *Grooms et al., 2023*; *Schweizer et al., 2022*).

Prior to this study, two systematic reviews published in 2014 (*Almonroeder, Garcia & Kurt, 2015*) and 2015 (*Brown, Brughelli & Hume, 2014*) investigated the effect of task anticipation (planned and unplanned) on lower-limb movement mechanics during single-limb changes of direction (sidestepping) in healthy individuals. Although these reviews were not focused on jump-landing tasks and were restricted to a specific type of task constraint, they suggest a possible link between task uncertainty and injury risk, as unplanned movements promoted knee mechanics that may increase the risk of injury (*Brown, Brughelli & Hume, 2014*; *Ness et al., 2020*). Additionally, another review in 2020 examined the evidence regarding the influence of dual-task (DT) conditions in individuals with anterior cruciate ligament (ACL) injuries (*Ness et al., 2020*). While the authors highlighted the usefulness of DT tests for assessing injured athletes, they also acknowledged the need for more sport-specific tests (such as cutting or jumping), as their review primarily covered studies using DT in static standing and walking tests. In recent years, several primary studies have been published on this topic. However, to date, no reviews have thoroughly analyzed the effect of incorporating DT during jumping or landing actions. Therefore, the objective of the present review is to examine the influence of performing a DT or sport-specific task constraints during jump-landing tests (vertical jump or horizontal jump) on biomechanical variables related to the risk of lower-limb injury in team sports.

## MATERIALS & METHODS

### Study design

This systematic review was registered in PROSPERO (registration number: CRD42023462102) and was conducted and reported following the Preferred Reporting Items for Systematic Reviews and Meta-Analyses (PRISMA) guidelines (*Page et al., 2021*). This review was intended to serve as a comprehensive evaluation of the impact of DT or sport-specific task constrains conditions on jumping and landing assessments, while also addressing their influence on biomechanical (Kinematics, *e.g.*, displacement, angles, *etc.* and kinetics, *e.g.*, forces and torques) risk factors associated with lower extremity injuries in the context of team sports.

### Data sources and search strategy

A systematic search was carried out using four electronic databases (MEDLINE (PubMed), Web of Science, Cochrane Plus and SportDiscus) for studies published from 2013 until June 30th, 2023. The search of the databases was limited to included articles published in the English language.

The specific search strategy for included databases was: (dual task OR task constraints OR ball OR cognitive task OR Virtual reality OR Attention OR Sports specific tasks) AND (jump OR Landing) AND (risk factor OR Biomechanics OR injury risk OR Motor control OR Movement variability) AND (sport).

### Eligibility criteria

To be eligible, studies had to include: (1) kinematic (*e.g.*, displacement, angles, *etc.*) and/or kinetic (*e.g.*, forces and torques) assessment of injury risk factors in the lower extremity: ankle, knee and/or hip; (2) A comparison between a simple jump or landing test and a DT jump or landing test which included cognitive information defined as arithmetic, auditory, visual and/or working memory (*Chaaban, Turner & Padua, 2023*); or alternatively, a comparison between two DT jump or landing tests of different difficulty.

The following studies were excluded: (1) studies that did not compare different levels of cognitive or coordinative demands on the jump or landing tasks; (2) studies without statistical analysis of kinematic and kinetic variables related to lower-limb injury risk; (3) studies whose aim and statistical calculations were focused on a different topic than the effects of adding a DT or constraint to jumping or landing actions and (4) studies in which more than 50% of the total number of participants do not participate in regular team sports competitions, and (5) studies that included populations with neurological conditions such as concussion or dementia.

### Study selection and data collection process

All identified references were downloaded from the databases to a unified Excel file, where duplicates were identified and removed. Three researchers (S.G-M, V.I-D and M.M-Y) carried out the review process following the inclusion/exclusion criteria. The process was conducted in three stages. In the first step, the authors independently screened the literature from titles, abstracts, and keywords after uploading the references to a management system,

Abstrackr (*Wallace et al., 2012*). In the second step, the full-text articles of relevant studies were examined independently by the aforementioned researchers to determine inclusion. In the third step, additional articles were searched for by one author (S.G-M) in the reference lists of all included articles. Any disagreement at any step of the process was discussed and resolved at a consensus meeting with a fourth author (B.F-V).

Study characteristics, encompassing publication details (such as authors, year, and study design), participants attributes (including participants, sample size, age, body mass, height, sex, sport and level of practice), protocol (jumping or landing task and DT conditions), measurement methodologies, and outcome measures (all results involving kinematic or kinetic variables), were systematically extracted and documented on a spreadsheet. The data extraction process was conducted by one author (S.G-M) and subjected to a rigorous validation procedure, with double-checking by a second author (V.I-D) to ensure accuracy and consistency.

## Classification of the dual-task constrains

The present review does not aim to establish rigid terminological or taxonomical foundations regarding the classification of dual-task (DT). Indeed, there is considerable terminological diversity in the scientific literature concerning the definition of what constitutes a DT (*McIsaac, Lamberg & Muratori, 2015*). According to the most commonly used taxonomy, a DT can be considered as the simultaneous performance of two tasks that can be conducted and assessed independently (*McIsaac, Lamberg & Muratori, 2015*; *Ness et al., 2020*). Moreover, the type of DT is commonly classified as Motor-Motor, in which two coordinative or motor tasks are combined, and Motor-Cognitive, in which a motor task is combined with cognitive processing (*McIsaac, Lamberg & Muratori, 2015*; *Mas, Naranjo & Mollá, 2023*; *Akin et al., 2021*; *O'Shea, Morris & Iansek, 2002*). It should be noted that cognitive processes are embodied with natural movement, as it requires attention and memory, such that each domain impacts the other (*Wilson, 2002*). Therefore, classification and taxonomy of the DT conditions is not intended to neatly categorize every specific combination of tasks. Rather, it is used to facilitate discussion and comparison with similar modalities. In this regard, motor tasks whose execution is conditioned by a stimulus, in which there is no need to decide between different response options (time conditioned—TC) or in which a choice is made between different responses depending on the stimulus (DM), are usually considered as cognitive conditioning factors (*Hughes & Dai, 2023*; *Chaaban, Turner & Padua, 2023*). However, since DM is one of the most specific cognitive factors in team sports, in the present review this factor has been treated and analyzed in the discussion section independently from other cognitive conditioning factors.

## Methodological quality

The included studies underwent evaluation using the quality assessment tool for cross-sectional and observational cohort studies proposed by the National Heart, Lung, and Blood Institute (NHLBI) (*Morral-Yepes et al., 2022*; *National Heart, Lung, and Blood Institute (NHLBI), 2021*; *Ullman, Fernandez & Klein, 2021*). No studies were excluded based on

their methodological quality. Owing to the methodological and statistical heterogeneity of the included studies, a descriptive approach was adopted in the research synthesis (*Rethlefsen et al., 2021*).

### Risk of bias assessment

The risk of bias in the selected articles was analyzed using the recommendations of the Cochrane Collaboration for systematic reviews (*Higgins et al., 2011*). The assessments of the researchers were classified as "low risk", "high risk", or "unclear risk of bias". The tool was comprised of the following items: random sequence generation, allocation concealment; blinding of participants and personnel, blinding of outcome assessment; incomplete outcome data; selective reporting, and other biases. Two researchers (SG-M and V.T-R) independently assessed the methodological quality of all selected articles. Another author resolved any potential discrepancies (V.I-D).

## RESULTS

### Selection of the studies

A total of 656 potentially relevant publications were identified as eligible from the selected databases. After screening the titles and abstracts, 610 (92.99%) were excluded for not meeting the inclusion criteria and 46 (7.01%) were selected for full-text review. One of the 46 potentially relevant publications (2.17%) could not be retrieved, therefore, 45 full-text articles were checked for eligibility by inclusion and exclusion criteria. After peer-review, 13 articles were included from database searching. Two additional articles were included *via* manually screened references from included articles and previous related systematic reviews. 15 studies were ultimately included for analysis (*Lin et al., 2020*; *Kajiwara et al., 2019*; *Stephenson et al., 2018*; *Wilke et al., 2021*; *Almonroeder et al., 2018*; *Akbari, Kuwano & Shimokochi, 2023*; *Fílter et al., 2022*; *Imai et al., 2022*; *Fischer et al., 2021*; *DiCesare et al., 2020*; *Alanazi et al., 2020*; *Beardt et al., 2018*; *Ren et al., 2022*; *Ford et al., 2017*; *Richwalski et al., 2018*) (Fig. 1). All the selected studies focus on jump-landing tests in team sports and biomechanics related to increased lower-limb injury risk.

### Quality assessment results

The quality assessment tool for cross-sectional and observational cohort studies by the NHLBI provides a maximum score of 14 points (*National Heart, Lung, and Blood Institute (NHLBI), 2021*). Nevertheless, due to the inherent characteristics of the scrutinized studies, certain criteria (questions 6, 7, 8, 10 and 13) were deemed not applicable (*Morral-Yepes et al., 2022*). Consequently, these specific questions were excluded from the scoring system, maintaining the highest possible score at 9 points. Within this framework, the studies included in the analysis achieved scores ranging from 5 to 7 points (see Table 1).

### Risk of bias

According to the Cochrane Collaboration recommendations, 12 studies did not surpass a score of 3 points (out of a total of 7, see Table 2). Only 3 studies exceeded a score of 3 points, with one article achieving a total score of 6. Three studies were found to have a

**Table 1** Evaluation of methodological quality according to the quality assessment tool for cross-sectional and observational cohort studies proposed by the *National Heart, Lung, and Blood Institute (NHLBI) (2021)*.

| Study, year | Q1 | Q2 | Q3 | Q4 | Q5 | Q6 | Q7 | Q8 | Q9 | Q10 | Q11 | Q12 | Q13 | Q14 | Total score (out of 9) | % |
|---|---|---|---|---|---|---|---|---|---|---|---|---|---|---|---|---|
| Akbari, Kuwano & Shimokochi (2023) | 1 | 1 | 0 | 1 | 0 | NA | NA | NA | 1 | NA | 1 | 0 | NA | 1 | 6 | 66.7 |
| Alanazi et al. (2020) | 1 | 1 | 0 | 1 | 1 | NA | NA | NA | 1 | NA | 1 | 0 | NA | 1 | 7 | 77.8 |
| Almonroeder et al. (2018) | 1 | 1 | 0 | 1 | 0 | NA | NA | NA | 1 | NA | 1 | 0 | NA | 1 | 6 | 66.7 |
| Beardt et al. (2018) | 1 | 1 | 0 | 1 | 1 | NA | NA | NA | 1 | NA | 1 | 0 | NA | 1 | 7 | 77.8 |
| DiCesare et al. (2020) | 1 | 1 | 0 | 1 | 0 | NA | NA | NA | 1 | NA | 1 | 0 | NA | 1 | 6 | 66.7 |
| Filter et al. (2022) | 1 | 1 | 0 | 1 | 0 | NA | NA | NA | 1 | NA | 1 | 0 | NA | 1 | 6 | 66.7 |
| Fischer et al. (2021) | 1 | 1 | 0 | 1 | 1 | NA | NA | NA | 1 | NA | 1 | 0 | NA | 1 | 7 | 77.8 |
| Ford et al. (2017) | 1 | 1 | 0 | 1 | 0 | NA | NA | NA | 1 | NA | 1 | 0 | NA | 1 | 6 | 66.7 |
| Imai et al. (2022) | 1 | 1 | 0 | 1 | 1 | NA | NA | NA | 1 | NA | 1 | 0 | NA | 1 | 7 | 77.8 |
| Kajiwara et al. (2019) | 1 | 1 | 0 | 1 | 0 | NA | NA | NA | 1 | NA | 1 | 0 | NA | 1 | 6 | 66.7 |
| Lin et al. (2020) | 1 | 0 | 0 | 1 | 0 | NA | NA | NA | 1 | NA | 1 | 0 | NA | 1 | 5 | 55.6 |
| Ren et al. (2022) | 1 | 1 | 0 | 1 | 1 | NA | NA | NA | 1 | NA | 1 | 0 | NA | 1 | 7 | 77.8 |
| Richwalski et al. (2018) | 1 | 1 | 0 | 1 | 1 | NA | NA | NA | 1 | NA | 1 | 0 | NA | 1 | 7 | 77.8 |
| Stephenson et al. (2018) | 1 | 1 | 0 | 1 | 1 | NA | NA | NA | 1 | NA | 1 | 0 | NA | 1 | 7 | 77.8 |
| Wilke et al. (2021) | 1 | 0 | 0 | 1 | 0 | NA | NA | NA | 1 | NA | 1 | 0 | NA | 1 | 5 | 55.6 |

**Notes.**

0, not fulfilled criterion; 1, fulfilled criterion; NA, not applicable; Q1, Was the research question or objective in this article clearly stated?; Q2, Was the study population clearly specified and defined?; Q3, Was the participation rate of eligible persons at least 50%?; Q4, Were all the subjects selected or recruited from the same or similar populations (including the same time period)? Were inclusion and exclusion criteria for being in the study prespecified and applied uniformly to all participants?; Q5, Was a sample size justification, power description, or variance and effect estimates provided?; Q6, For the analyses in this article, were the exposure(s) of interest measured prior to the outcome(s) being measured?; Q7, Was the timeframe sufficient so that one could reasonably expect to see an association between exposure and outcome if it existed?; Q8, For exposures that can vary in amount or level, did the study examine different levels of the exposure as related to the outcome (*e.g.*, categories of exposure, or exposure measured as continuous variable)?; Q9, Were the exposure measures (independent variables) clearly defined, valid, reliable, and implemented consistently across all study participants?; Q10, Was the exposure(s) assessed more than once over time?; Q11, Were the outcome measures (dependent variables) clearly defined, valid, reliable, and implemented consistently across all study participants?; Q12, Were the outcome assessors blinded to the exposure status of participants?; Q13, Was loss to follow-up after baseline 20% or less?; Q14, Were key potential confounding variables measured and adjusted statistically for their impact on the relationship between exposure(s) and outcome(s)?.

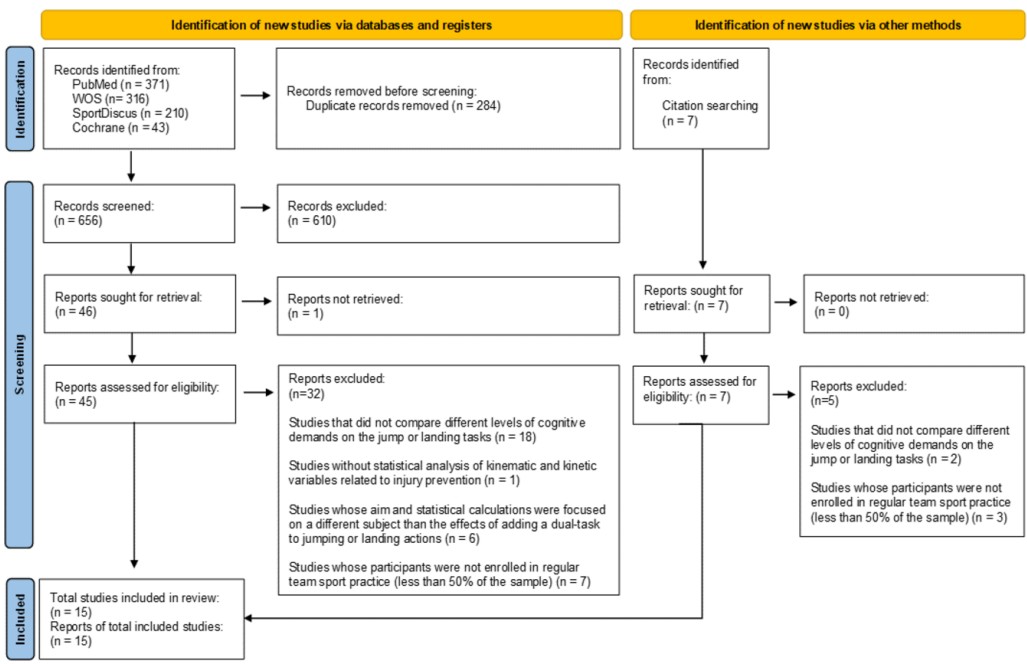

**Figure 1** Flowchart representing the identification and selection processes of relevant studies according to the PRISMA 2020 declaration (*Chaaban, Turner & Padua, 2023*).

high risk of bias in random sequence generation. Furthermore, all studies exhibited a high risk of bias in allocation concealment. On the other hand, all studies were assessed to have a low risk of bias in selective reporting. A single study reported a low risk of bias in the blinding of participants, personnel, and outcome assessment. Lastly, two studies had an unclear risk of bias in incomplete outcome data.

## Participant's characteristics

Table 3 provides a comprehensive overview of the key characteristics pertaining to the participants involved in the selected studies. Across the various studies, a total of 347 individuals were subjected to analysis, comprising 68 males, 249 females, and 30 individuals for whom sex or gender information was not specified. The participant count in each study ranged from 12 to 40 individuals. Noteworthy is that, within the 15 studies examined, nine exclusively involved female participants (*Wilke et al., 2021*; *Almonroeder et al., 2018*; *Fílter et al., 2022*; *Imai et al., 2022*; *Fischer et al., 2021*; *DiCesare et al., 2020*; *Beardt et al., 2018*; *Ford et al., 2017*; *Richwalski et al., 2018*), whereas one study comprised solely male participants (*Ren et al., 2022*).

The average age of participants spanned from 16.0 to 26.1 across the studies, with one study specifically involving adolescents (*DiCesare et al., 2020*) and the remaining 14 focusing on young adults (*Lin et al., 2020*; *Kajiwara et al., 2019*; *Stephenson et al., 2018*; *Wilke et al., 2021*; *Almonroeder et al., 2018*; *Akbari, Kuwano & Shimokochi, 2023*; *Fílter et al., 2022*; *Imai et al., 2022*; *Fischer et al., 2021*; *Alanazi et al., 2020*; *Beardt et al., 2018*; *Ren et al., 2022*; *Ford et al., 2017*; *Richwalski et al., 2018*). Concerning athletic engagement, the

**Table 2  Results of the risk of bias assessment of included studies.**

| Study, years | Items | | | | | | | Total (out of 7 items) | % |
|---|---|---|---|---|---|---|---|---|---|
| | 1 | 2 | 3 | 4 | 5 | 6 | 7 | | |
| *Akbari, Kuwano & Shimokochi (2023)* | X | X | X | X | X | ✓ | ✓ | 2 | 28.5 |
| *Alanazi et al. (2020)* | ✓ | X | X | X | X | ✓ | ✓ | 3 | 42.8 |
| *Almonroeder et al. (2018)* | ✓ | X | X | X | X | ✓ | ✓ | 3 | 42.8 |
| *Beardt et al. (2018)* | ✓ | X | X | X | ¿? | ✓ | ✓ | 3 | 42.8 |
| *DiCesare et al. (2020)* | X | X | X | X | ✓ | ✓ | ✓ | 3 | 42.8 |
| *Fílter et al. (2022)* | ✓ | X | X | X | X | ✓ | ✓ | 3 | 42.8 |
| *Fischer et al. (2021)* | ✓ | X | X | X | ✓ | ✓ | ✓ | 4 | 57.1 |
| *Ford et al. (2017)* | ✓ | X | X | X | X | ✓ | ✓ | 3 | 42.8 |
| *Imai et al. (2022)* | X | X | X | X | X | ✓ | ✓ | 2 | 28.5 |
| *Kajiwara et al. (2019)* | ✓ | X | X | X | ¿? | ✓ | ✓ | 3 | 42.8 |
| *Lin et al. (2020)* | ✓ | X | X | X | X | ✓ | ✓ | 3 | 42.8 |
| *Ren et al. (2022)* | ✓ | X | X | X | X | ✓ | ✓ | 3 | 42.8 |
| *Richwalski et al. (2018)* | ✓ | X | X | X | X | ✓ | ✓ | 3 | 42.8 |
| *Stephenson et al. (2018)* | ✓ | X | ✓ | ✓ | ✓ | ✓ | ✓ | 6 | 85.7 |
| *Wilke et al. (2021)* | ✓ | X | X | X | ✓ | ✓ | ✓ | 4 | 57.1 |
| Total out of 15 articles (%) | 12 (80.0%) | 0 (0.0%) | 1 (6.6%) | 1 (6.6%) | 4 (26.6%) | 15 (100%) | 15 (100%) | | |

**Notes.**

1, Random sequence generation; 2, Allocation concealment; 3, Blinding of participants and personnel; 4, Blinding of outcome assessment; 5, Incomplete outcome data; 6, Selective reporting; 7, Other bias; ✓, low risk of bias; X, high risk of bias; ¿?, unclear risk of bias; the higher the score, the lower the risk of bias.

included investigations involved a heterogeneous cohort of individuals participating in diverse team sports, such as basketball, volleyball, soccer, and handball. The participant pool covered a spectrum of skill levels, incorporating recreational athletes (*Stephenson et al., 2018*; *Fischer et al., 2021*; *Alanazi et al., 2020*), individuals from high school, college, and university settings (*Lin et al., 2020*; *Kajiwara et al., 2019*; *Wilke et al., 2021*; *Almonroeder et al., 2018*; *Akbari, Kuwano & Shimokochi, 2023*; *Imai et al., 2022*; *DiCesare et al., 2020*; *Beardt et al., 2018*; *Ford et al., 2017*; *Richwalski et al., 2018*), and semi-professional and professional athletes (*Fílter et al., 2022*; *Ren et al., 2022*). Except for one study, all the investigations exclusively focused on healthy volunteers. The outlier study sought to explore distinctions between healthy volunteers and participants who had undergone ACL reconstruction and rehabilitation (*Alanazi et al., 2020*).

## Jump-Landing protocols and dual tasks conditions

Table 4 summarizes the principal characteristics of the study protocols included in the review. As shown, most of the studies employ the DJ as the standard (control) test to benchmark against one or multiple variations of the jump or DT conditions. The comparative conditions comprise actions influenced by the inclusion of a sports ball (six
**Table 3 Participant's characteristics.**

| Study, year | n | Sex F | Sex M | Age (y) | Height (m) | Weight (kg) | Sport Soccer | Basketball | Voleyball | Handball | Other Sports | Level of practice | Training volume | Medical history (Injured/ Uninjured) |
|---|---|---|---|---|---|---|---|---|---|---|---|---|---|---|
| Akbari, Kuwano & Shimokochi (2023) | 24 | 18 | 6 | 20.04 ±1.12 | 1.66 ±0.07 | 61.0 ±8.5 | ● | | | | | ○○③○ | NA | Uninjured |
| Alanazi et al. (2020) | 36 | 20 | 16 | ACLR: 26.11 ±3.95 | ACLR: 1.70 ±0.09 | ACLR: 68.2 ±9.6 | ● | | | | | ①○○○ | NA | 18 Uninjured (F = 10; M = 8) |
| | | | | CON: 25.83 ±3.51 | CON: 1.66 ±0.05 | CON: 66.9 ±0.4 | | | | | | | | 18 ACLR (F = 10; M = 8) |
| Almonroeder et al. (2018) | 20 | 20 | | 21.5 ±1.8 | 1.70 ±0.10 | 64.1 ±1.2 | | ● | | | | ○②○○ | Tegner Activity Level scale score >4/10 | Uninjured |
| Beardt et al. (2018) | 17 | 17 | | 20.0 ±1.7 | 1.68 ±0.07 | 65.9 ±9.9 | | | ● | | | ○②③○ | NA | Uninjured |
| DiCesare et al. (2020) | 38 | 38 | | 16.0 ±1.3 | 1.65 ±0.06 | 59.5 ±9.9 | ● | | | | | ○○③○ | NA | Uninjured |
| Filter et al. (2022) | 12 | 12 | | 23.9 ±3.5 | 1.75 ±0.05 | 71.6 ±3.5 | ● | | | | | ○○○④ | 5–6 training sessions & 1–2 matches/week. | Uninjured |
| Fischer et al. (2021) | 40 | 40 | | 20.2 ±2.6 | 1.69 ±0.07 | 64.1 ±8.3 | ● | ● | | | | ①○○○ | 3 times/week | Uninjured |
| Ford et al. (2017) | 14 | 14 | | 18.8 ±1.1 | 1.63 ±0.08 | 63.0 ±7.9 | ● | ● | | | ● | ○②○○ | NA | Uninjured |
| Imai et al. (2022) | 20 | 20 | | 20.2 ±1.3 | NA | NA | ● | ● | ● | | ● | ○○③○ | Tegner Activity Level scale score >7/10 | Uninjured |
| Kajiwara et al. (2019) | 20 | 10 | 10 | 20.0 ±1.1 | 1.67 ±0.10 | 64.0 ±8.8 | ● | | | ● | | ○○③○ | NA | Uninjured |
| Lin et al. (2020) | 30 | NA | NA | 20.0 ±2.0 | 1.76 ±0.08 | 68.9 ±9.0 | | ● | ● | | ● | ○○③○ | NA | Uninjured |
| Ren et al. (2022) | 15 | | 15 | 20.1 ±1.5 | 1.81 ±0.07 | 75.4 ±10.7 | ● | | | | | ○○○④ | NA | Uninjured |
| Richwalski et al. (2018) | 12 | 12 | | 20.0 ±1.3 | 1.70 ±0.04 | 70.8 ±10.4 | | | ● | | | ○○③○ | NA | Uninjured |
| Stephenson et al. (2018) | 34 | 14 | 20 | 21.7 ±3.2 | 1.75 ±0.09 | 71.7 ±13.3 | ● | ● | ● | | ● | ①○○○ | 2 times/week, total of >3 h | Uninjured |
| Wilke et al. (2021) | 15 | 15 | | 25.8 ±0.4 | 1.71 ±0.07 | 68.0 ±12.0 | ● | ● | | ● | | ○○③○ | 8 ±3 h/week | Uninjured |

**Notes.**

Age, height and weight data are shown as mean ± standard deviation.

Level of practice: 1, Recreational; 2, High School; 3, College and University; 4, Semi-professional and Professional; ACLR, anterior cruciate ligament reconstruction; NA, Not available; F, Female; M, Male.

studies (*Almonroeder et al., 2018*; *Akbari, Kuwano & Shimokochi, 2023*; *Fílter et al., 2022*; *DiCesare et al., 2020*; *Alanazi et al., 2020*; *Beardt et al., 2018*)), performance of the task in virtual environments or with virtual feedback (two studies (*DiCesare et al., 2020*; *Ford et al., 2017*)), engagement in cognitive tasks (memory, two studies (*Wilke et al., 2021*; *Fischer et al., 2021*); arithmetic, one study (*Imai et al., 2022*); perceptive three studies (*Kajiwara et al., 2019*; *Ren et al., 2022*; *Ford et al., 2017*)) and tasks involving DM processes (seven studies (*Lin et al., 2020*; *Kajiwara et al., 2019*; *Stephenson et al., 2018*; *Almonroeder et al., 2018*; *Fischer et al., 2021*; *Beardt et al., 2018*; *Richwalski et al., 2018*)).

### Outcome assessments

Table 5 provides a comprehensive synthesis of the empirical evidence collated from the registries included in the systematic review. The table delineates the assessment instruments and methodologies employed across the selected studies to evaluate the outcomes. Diverse metrics were quantified, encompassing kinematic parameters such as lower-limb angles, kinetic variables including forces and torques, and additional indices such as stability and time to stabilization (TTS). These measures are intrinsically linked to the risk factors associated with lower extremity injuries within team sports contexts.

## DISCUSSION

The present article addresses a review of the effects of incorporating DT or sport-specific task constrains during the jumping or landing action, both coordinative (motor-motor), cognitive (motor-cognitive) or conditioned by DM, on the biomechanical variables linked to lower limb injury risk factors in team sports. The reviewed studies demonstrate that the inclusion of coordinative DT, cognitive DT, or tasks involving DM in jump or landing actions can significantly alter various biomechanical variables such as joint torques, as well as variations in stability and postural control, which are directly related to the risk of lower limb injuries in team sports. This section has been organized based on the classification of the dual-task (DT) constrains presented in the materials & methods section. Thus, in the following paragraphs, the effects of coordinative DT, cognitive DT, or tasks involving DM on jump-landing tests are analysed.

Regarding studies that evaluated the effects of introducing constraints or coordinative DT (motor-motor) (*Almonroeder et al., 2018*; *Akbari, Kuwano & Shimokochi, 2023*; *Fílter et al., 2022*; *DiCesare et al., 2020*; *Alanazi et al., 2020*; *Beardt et al., 2018*; *Ford et al., 2017*), significant changes in biomechanical parameters related to injury risk were recorded in all of them. From a kinetic perspective, increases in peak vertical ground reaction forces (vGRF) during landing (*Almonroeder et al., 2018*), rate of force development, and mechanical impulse (*Fílter et al., 2022*), along with increases in peak knee abduction torque (*Almonroeder et al., 2018*), were reported. From a kinematic perspective, a decrease in hip, knee, and ankle flexion, hip abduction, and ankle inversion at initial contact was observed (*Akbari, Kuwano & Shimokochi, 2023*; *DiCesare et al., 2020*; *Alanazi et al., 2020*; *Beardt et al., 2018*), along with an increase in temporal differences in initial contact between both feet (*Beardt et al., 2018*). Additionally, the peak knee flexion angle (*Almonroeder et al., 2018*; *Akbari, Kuwano & Shimokochi, 2023*), vertical

González-Millán et al. (2024), *PeerJ*, DOI 10.7717/peerj.17720

**Table 4 Jump-landing protocols and dual tasks or constrained conditions.**

| Study, year | CONDITIONS | Details of the jump-landing tests | Proposed dual task conditions | Motor–cognitive tasks | | | Recall | Arithmetic | 0thers | Motor–motor tasks |
|---|---|---|---|---|---|---|---|---|---|---|
| | | | | Time conditioned actions/Response to stimuli | | | | | | |
| | | | | NTC | DM | Stimuli | | | | |
| *Akbari, Kuwano & Shimokochi (2023)* | **DVJ** *(Control)* | **DVJ:** Jump off the box (30 cm), maximum jump with both feet and landing with a single foot, arms free. **DVJ + ball:** Same as DVJ plus heading a suspended soccer ball. | **DVJ + ball:** | – | – | – | – | – | – | Soccer head |
| | **DVJ + ball** | **Number of repetitions:** Three successful trials of each condition. (Total: 6) | | | | | | | | |
| | | **Intervals:** Sufficient rest between trials for full recovery (self-determined) | | | | | | | | |
| *Alanazi et al. (2020)* | **FJ** *(Control)* | **FJ:** Bilateral jump forward (80% of maximum long jump) and landing with both feet, arms free. **FJ + ball:** Same as FJ plus heading a suspended soccer ball. | **FJ + ball:** | – | – | – | – | – | – | Soccer head |
| | **FJ + ball** | **Number of repetitions:** Four successful trials of each condition. (Total: 8) | | | | | | | | |
| | | **Intervals:** NA. | | | | | | | | |
| *Almonroeder et al. (2018)* | **DVJ** *(Control)* | **DVJ:** Jump off the box (31 cm), maximum jump with both feet and landing with both feet, arms free. **DVJ + ball:** Same as DJV plus grabbing an overhead suspended basketball with both hands. **DVJ DM:** Same as DJV but a screen illuminated one of two possible lights approximately 250 ms before first ground contact to indicate jump or just land (only actual jumps were analyzed). **DVJ DM + ball:** A combination of the two previous conditions. | **DVJ + ball** | – | – | – | – | – | – | Basketball grab (hands) |

**Table 4** (*continued*)

| Study, year | CONDITIONS | Details of the jump-landing tests | Proposed dual task conditions | Motor–cognitive tasks | | | | | | Motor–motor tasks |
|---|---|---|---|---|---|---|---|---|---|---|
| | | | | Time conditioned actions/Response to stimuli | | | Recall | Arithmetic | Others | |
| | | | | NTC | DM | Stimuli | | | | |
| | DVJ + ball | **Number of repetitions:** Three successful trials of each condition. (Total: 12) | **DVJ DM** | – | ● | Two lights on screen | – | – | – | – |
| | DVJ DM | **Intervals:** Sufficient rest between trials for full recovery (self-determined) | **DVJ DM + ball** | – | ● | Two lights on screen | – | – | – | Basketball grab (hands) |
| | DVJ DM + ball | | | | | | | | | |
| *Beardt et al. (2018)* | DVJ (Control) | **DVJ:** Jump off the box (30 cm), maximum jump with both feet and landing with both feet, arms free. **Volley:** Approach to a volleyball net, jump with both feet, spike a ball that has been tossed by a player. **Volley DM:** Same as Volleyball but trying to pass two front-row players on an opposing team trying to block. | **Volley** | ● | – | Ball pass | – | – | – | Volleyball spike (hand) |
| | Volley | **Number of repetitions:** Five successful trials of each condition. (Total: 15) | **Volley DM** | – | ● | Ball pass + opponents | – | – | – | Volleyball Spike (hand) |
| | Volley DM | **Intervals:** NA. | | | | | | | | |
| *DiCesare et al. (2020)* | DVJ + ball (Control) | **DVJ + ball:** Jump off the box (31 cm), maximum jump with both feet and landing with both feet, arms free, reaching a suspended ball with both hands. **VR soccer:** Jump and head a virtual soccer ball that was kicked by a computer-controlled player (VR). | **DVJ + ball** | – | – | – | – | – | – | Soccer grab (hands) |
| | VR soccer | **Number of repetitions:** Three successful trials of DVJ + ball and two practice trials followed by four successful trials of VR soccer. (Total: 9) | **VR soccer** | ● | – | Ball pass VR | – | – | – | Soccer (VR) head |
| | | **Intervals:** NA. | | | | | | | | |

Peer J

**Table 4** (*continued*)

| Study, year | Jump-landing TESTS (task protocols) | | Proposed dual task conditions | Dual task/constraints characteristics | | | | | | Motor–motor tasks |
|---|---|---|---|---|---|---|---|---|---|---|
| | | | | Motor–cognitive tasks | | | | | | |
| | CONDITIONS | Details of the jump-landing tests | | Time conditioned actions/Response to stimuli | | | Recall | Arithmetic | Others | |
| | | | | NTC | DM | Stimuli | | | | |
| *Fílter et al. (2022)* | **Run up VJ** *(Control)* | **Run up VJ**: Run 5.5 m (approach), maximum VJ with both feet and landing with both feet, arms free. **Run up VJ + ball**: Same as Run up VJ plus heading a suspended soccer ball, jump was not forced to be bilateral. | **Run up VJ + ball:** | – | – | – | – | – | – | Soccer head |
| | **Run up VJ + ball:** | **Number of repetitions:** Three successful trials of each condition. (Total: 6) | | | | | | | | |
| | | **Intervals:** 60-second rest between trials and 2-minute rest between conditions. | | | | | | | | |
| *Fischer et al. (2021)* | **DVJ** *(Control)* | **DVJ:** Jump off the box (30 cm), maximum jump with both feet and landing with both feet on specific areas (A or B) 1 m forward and 45° to the right or left, arms free. **DVJ DM:** Same as DVJ but an arrow displayed on a screen (left or right) approximately 250 ms before first ground contact indicated jump to A or B specific area. **DVJ Recall:** Same as DVJ but before starting, six numbers were presented simultaneously on different positions of the screen for 1 s. After landing, participants were asked for the number of one of the six possible positions. **DVJ Recall + Attention:** Same as DJV Recall, but the position of the number to recall was shown on the screen approximately 250 ms before first ground contact for 1 s to force attentional focus. **DVJ Recall + Attention DM:** A combination of DJV Recall + Attention with DM. | **DVJ DM** | – | • | Arrows on screen | – | – | – | – |
| | **DVJ DM** | **Number of repetitions:** Three successful trials of each direction for each condition. (Total: 30) | **DVJ Recall** | – | – | | Numbers on screen | – | – | – |
| | **DVJ Recall** | **Intervals:** NA. | **DVJ Recall + Attention** | – | – | | Numbers on screen | – | – | – |

González-Millán et al. (2024), *PeerJ*, DOI 10.7717/peerj.17720

**Table 4** (*continued*)

| Study, year | Jump-landing TESTS (task protocols) | | Dual task/constraints characteristics | | | | | | |
|---|---|---|---|---|---|---|---|---|---|
| | CONDITIONS | Details of the jump-landing tests | Proposed dual task conditions | Motor–cognitive tasks | | | | | Motor–motor tasks |
| | | | | Time conditioned actions/Response to stimuli | | | Recall | Arithmetic 0thers | |
| | | | | NTC | DM | Stimuli | | | |
| | DVJ Recall + Atention | | DVJ Recall + Attention DM | – | ● | Arrows on screen | Numbers on screen | – – | – |
| | DVJ Recall + Attention DM | | | | | | | | |
| *Ford et al. (2017)* | DVJ + Overhead goal *(Control)* | **DVJ + Overhead goal:** Jump off the box (31 cm), maximum jump with both feet and landing with both feet, arms free, reaching an overhead goal with both hands. **DVJ + Virtual Overhead goal:** Same DVJ but participants were focused on a Virtual 3D biomechanical model of themselves on a screen and the target was an overhead virtual target. | DVJ + Virtual Ball | – | – | – | – | – Perception of 3D body-ball model | Reach virtual overhead goal (hands) |
| | DVJ + Virtual Overhead goal | **Number of repetitions:** Three successful trials of each condition. (Total: 6) | | | | | | | |
| | | **Intervals:** NA. | | | | | | | |
| *Imai et al. (2022)* | DVJ *(Control)* | **DVJ:** Jump off the box (30 cm), maximum jump with both feet and landing with both feet, arms free. **DVJ + calculation:** Same as DJV but a 2-digit sum was displayed on a screen just before the DVJ. Thereafter each participant gave the answer after the DVJ. | DVJ + calculation | – | – | – | – | Addition of 2-digit – | – |
| | DVJ + calculation | **Number of repetitions:** Three successful trials of DVJ and two successful trials of DVJ + calculation. (Total: 5) | | | | | | | |
| | | **Intervals:** 3-minute rest between conditions. | | | | | | | |

González-Millán et al. (2024), *PeerJ*, DOI 10.7717/peerj.17720

**Table 4** (*continued*)

| Study, year | Jump-landing TESTS (task protocols) | | Dual task/constraints characteristics | | | | | | | |
|---|---|---|---|---|---|---|---|---|---|---|
| | **CONDITIONS** | **Details of the jump-landing tests** | **Proposed dual task conditions** | **Motor–cognitive tasks** | | | | | | **Motor–motor tasks** |
| | | | | **Time conditioned actions/Response to stimuli** | | | **Recall** | **Arithmetic** | **Others** | |
| | | | | NTC | DM | Stimuli | | | | |
| *Kajiwara et al. (2019)* | **SDL** (*Control*). | **SDL:** Jump off the box (30 cm) and land on one foot remaining stationary for 2 s, keeping hands on the hips. **SDL Stroop test:** Same as SDL but the words "blue", "red", or "yellow" were displayed on a screen in a font color different to the meaning of the word. Subjects were told to respond to the color of the text, not its meaning, landing on a specific site on the floor depending on the color. | **SDL Stroop test** | – | ● | Word + color on screen (3 options) | – | – | Stroop test (discriminate word) | – |
| | **SDL Stroop test** | **Number of repetitions:** Three successful trials of each condition. (Total: 6) | | | | | | | | |
| | | **Intervals:** Sufficient rest between trials for full recovery (self-determined) | | | | | | | | |
| *Lin et al. (2020)* | **DVJ + Overhead goal** (*Control*). | **DVJ + Overhead goal**: Jump off the box (30 cm), maximum jump with both feet to touch and overhead target with the dominant hand and land on the dominant foot remaining stable for 5 s. **DVJ + Overhead goal DM:** Same as DVJ + Overhead goal but the overhead goal were three reaction lights placed above in left, center, and right positions. The target light was illuminated with first ground contact. | **DVJ + Overhead goal** | – | – | – | – | – | – | Reach overhead goal (dominant hand) |
| | **DVJ + Overhead goal DM** | **Number of repetitions:** Three successful trials of each condition. (Total: 6) | **DVJ + Overhead goal DM** | – | ● | 3 reaction lights | – | – | – | Reach overhead goal (dominant hand) |
| | | **Intervals:** 60-second rest between trials and 5-minute rest between conditions. | | | | | | | | |

Gonzáles-Millán et al. (2024), *PeerJ*, DOI 10.7717/peerj.17720

**Table 4** (*continued*)

| Study, year | Jump-landing TESTS (task protocols) | | Dual task/constraints characteristics | | | | | | | |
|---|---|---|---|---|---|---|---|---|---|---|
| | CONDITIONS | Details of the jump-landing tests | Proposed dual task conditions | Motor–cognitive tasks | | | | | | Motor–motor tasks |
| | | | | Time conditioned actions/Response to stimuli | | | Recall | Arithmetic | Others | |
| | | | | NTC | DM | Stimuli | | | | |
| *Ren et al. (2022)* | DL (*Control*) | **DL:** Step forward from a box (40 cm), landing on both feet. **DL visual-cognitive**: Same as DL but eight spheres were displayed on a screen and participants must pay attention to one of them. Spheres start to move and interact in space as participants perform the DL. At the end of the task, the participants point out the position of the target sphere on the screen. | DL visual-cognitive | – | – | – | – | – | Follow an object trajectory and interactions | – |
| | DL visual-cognitive | **Number of repetitions:** NA. | | | | | | | | |
| | | **Intervals:** NA. | | | | | | | | |
| *Richwalski et al. (2018)* | DL side/Jump (*Control*) | **DL side/jump:** Jump off the box (40 cm), landing on both feet. Immediately following landing, participants jumped straight up, or cross stepped laterally to a target placed 1.5 m away, in accordance with a directional arrow (Right/Left/Up) displayed on screen before starting the test. **DL side/Jump DM**: Same as DL side/jump but arrows were triggered by participants motion while stepping of the box. | DL side/jump DM | – | ● | Arrows on screen | – | – | – | – |
| | DL side/Jump DM | **Number of repetitions:** Five trials for each condition and direction. (Total: 30) | | | | | | | | |
| | | **Intervals:** 30-to-45-second rest between trials. | | | | | | | | |

Gonzàlez-Millán et al. (2024), *PeerJ*, DOI 10.7717/peerj.17720

**Table 4** (*continued*)

| Study, year | Jump-landing TESTS (task protocols) | | Dual task/constraints characteristics | | | | | | | |
|---|---|---|---|---|---|---|---|---|---|---|
| | CONDITIONS | Details of the jump-landing tests | Proposed dual task conditions | Motor–cognitive tasks | | | | | | Motor–motor tasks |
| | | | | Time conditioned actions/Response to stimuli | | | Recall | Arithmetic | 0thers | |
| | | | | NTC | DM | Stimuli | | | | |
| *Stephenson et al. (2018)* | **DL side/Jump** (*Control*) | **DL side/jump:** Jump off the box (30 cm), landing on both feet. Immediately following landing, participants jumped straight up, or a 90° jump to the right or left in accordance with a directional visual stimulus (light) illuminated before starting the test. **DL side/Jump DM**: Same as DL side/jump but the visual stimuli were triggered by participants motion while stepping of the box. **DL side/Jump DM300**: Same as DL side/jump but the visual stimuli were activated 300 ms before landing. **DL side/Jump DM150**: Same as DL side/jump but the visual stimuli were activated 150 ms before landing. **DL side/Jump DMlanding**: Same as DL side/jump but the visual stimuli were activated simultaneously with landing. | **DL side/Jump DM** | – | ● | Directional lights | – | – | – | – |
| | **DL side/Jump DM** | **Number of repetitions:** Three trials for each condition and direction. (Total 45) | **DL side/Jump DM300** | – | ● | Directional lights | – | – | – | – |
| | **DL side/Jump DM300** | **Intervals:** 30-second rest between trials. | **DL side/Jump DM150** | – | ● | Directional lights | – | – | – | – |
| | **DL side/Jump DM150** | | **DL side/Jump DMland-ing** | – | ● | Directional lights | – | – | – | – |
| | **DL side/Jump DMlanding** | | | | | | | | | |

González-Millán et al. (2024), *PeerJ*, DOI 10.7717/peerj.17720

**Table 4** (*continued*)

| Study, year | CONDITIONS | Details of the jump-landing tests | Proposed dual task conditions | Motor–cognitive tasks | | | Recall | Arithmetic | Others | Motor–motor tasks |
|---|---|---|---|---|---|---|---|---|---|---|
| | | | | Time conditioned actions/Response to stimuli | | | | | | |
| | | | | NTC | DM | Stimuli | | | | |
| *Wilke et al. (2021)* | **CMJ number recall 1** | **CMJ:** Bilateral countermovement jump from the ground and land on one foot remaining stationary for 20 s. **CMJ number recall 1:** Photo depicting a typical game situation in American Football and was briefly (150 ms) shown during the flight phase. While the photo remained the same, depending on the condition, either one (N1), two (N2), or three (N3) shirt numbers (single digits) of the depicted players were presented. | **CMJ number recall 1** | – | – | – | 1 number | – | – | – |
| | **CMJ number recall 2** | **Number of repetitions:** Ten successful trials of each condition. (Total: 30) **Intervals:** The 30 jumps were split into blocks of six (two of each condition) with 2-minute rest intervals between blocks. | **CMJ number recall 2** | – | – | – | 2 numbers | – | – | – |
| | **CMJ number recall 3** | | **CMJ number recall 3** | – | – | – | 3 numbers | – | – | – |

**Notes.**

Classification of dual task characteristics adapted from *McIsaac, Lamberg & Muratori (2015)* and *Mas, Naranjo & Mollá (2023)*.

DL, Drop landing; DM, Decision Making; DVJ, Drop Vertical Jump; FJ, Forward Jump; NTC, Not time-conditioned (free start); SDL, Single leg drop landing; STC, Simple time conditioned; VJ, Vertical Jump; VR, Virtual Reality; NA, Not Available; ms, milliseconds.

**Table 5  Main outcomes of the included studies.**

| Author, year | Testing conditions | Performance/ errors in secondary task | Assessment instruments (analysis systems) | Variables | Main findings |
|---|---|---|---|---|---|
| *Akbari, Kuwano & Shimokochi (2023)* | **DVJ** *(Control)* | NA | **3D electromagnetic motion-tracking system** | **Kinematic variables**: Lower limb angles and angular displacements, COM | **DVJ + ball** showed ↓ peak knee flexion angle, knee flexion displacement, and center of mass vertical displacement. |
| | **DVJ + ball** | | Ascension Star (Ascension Technology, Burlington, VT, USA) | **Kinetic variables**: Lower limb joint moments, vGRF. | **DVJ + ball** showed ↑ knee joint stiffness. |
| | | | Frequency: 140 Hz; Markers: 4 | Kinetic data were normalized by body mass. | |
| | | | **Force platforms (x2)** | **Others**: Segmental and articular stiffness. | |
| | | | Bertec 4060 (Bertec, Columbus, OH, USA) | | |
| | | | Frequency: 1400 Hz; Axes: triaxial | | |
| *Alanazi et al. (2020)* | **FJ** *(Control)* | NA | **3D motion-tracking opto-electronic camera system (10-camera)** | **Kinematic variables**: Peak lower limb angles. | **FJ + ball** showed ↓ in hip flexion, knee flexion, hip extension moments, knee extension moments and peak pressure. |
| | **FJ + ball** | | Vicon Motion Systems (Vicon, Denver, CO, USA) | **Kinetic variables**: Peak lower limb moments and peak pressure. Joint moments were normalized by body mass. | |
| | | | Frequency: 240 Hz; Markers: 15 retro-reflective | **Others**: %MVC of lower limb muscles. | |
| | | | **Force platforms (x4)** | | |
| | | | AMTI (Advanced Mechanical Technology, Watertown, MA, USA) | | |
| | | | Frequency: 1920 Hz; Axes: triaxial | | |
| | | | **Plantar pressure measurement system (shoe insoles)** | | |
| | | | F-Scan wireless system (Tekscan, Boston, MA, USA) | | |
| | | | Frequency: 30 Hz | | |
| | | | **EMG (x14 wireless electrodes)** | | |
| | | | Trigno Wireless EMG (Delsys, Boston, MA, USA) | | |
| | | | Bandwidth, 20–450 Hz | | |

| Author, year | Testing conditions | Performance/ errors in secondary task | Assessment instruments (analysis systems) | Variables | Main findings |
|---|---|---|---|---|---|
| *Almonroeder et al. (2018)* | DVJ *(Control)* | Number of attempts (ranged from 15 to 26 trials) | **3D motion-tracking opto-electronic camera system (10-camera)** | **Kinematic variables**: Knee angles, ST. | **DVJ + ball** showed ↓ ST compared to **DVJ** and **DVJ DM + ball**. |
| | DVJ + ball | | Eagle system (Motion Analysis, Santa Rosa, CA, USA) | **Kinetic variables**: Knee moments, vGRF. | **DVJ + ball** showed ↑ peak vGRFs compared to the **DVJ**. |
| | DVJ DM | | Frequency: 200 Hz; Markers: 26 retro-reflective | Kinetic data were normalized by body mass. | **DVJ + ball** and **DVJ DM + ball** demonstrated ↓ peak knee flexion angles compared to the **DVJ**. |
| | DVJ DM + ball | | **Force platforms (x2)** | | **DVJ + ball** showed ↓ peak knee flexion angles compared to the **DVJ DM**. |
| | | | Bertec 4060 (Bertec, Columbus, OH, USA) | | **DVJ + ball** and **DVJ DM + ball** showed ↑ peak knee abduction angles compared to **DVJ**. |
| | | | Frequency: 1000 Hz; Axes: tri-axial | | |
| *Beardt et al. (2018)* | DVJ *(Control)* | NA | **Digital camcorders (x3: frontal and both sides of sagittal plane)** | **Jump performance**: Jump height | **Voley** and **Voley DM** showed ↑ in jump height, time differences in initial contact between two feet and knee and hip flexion at initial contact compared **DVJ**. |
| | Volley | | JVC GC-PX10 (JVC, Tokyo, Japan) | **Kinematic variables**: Lower limb angles, ST. | **Voley** and **Voley DM** showed ↓ ST, knee, and hip flexion for left and right leg and, knee-angle distance ratio compared **DVJ**. |
| | Volley DM | | Frequency: 60 Hz, Markers: 6 (tape to assist visual estimation of body landmarks for manual digitization) | | |
| *DiCesare et al. (2020)* | DVJ + ball *(Control)* | NA | **3D motion-tracking opto-electronic camera system (39-camera)** | **Kinematic variables**: Lower limb angles | **VR soccer** showed ↓ flexion at the hip and ankle at the point of peak knee flexion, hip abduction, and ankle inversion at initial contact. |
| | VR soccer | | (Motion Analysis, Santa Rosa, CA, USA) | | |
| | | | Frequency: 240 Hz for DVJ + ball and 120 Hz for VR soccer; | | |
| | | | Markers: retro-reflective (37 for DVJ + ball and 42 for VR soccer) | | |

| Author, year | Testing conditions | Performance/ errors in secondary task | Assessment instruments (analysis systems) | Variables | Main findings |
|---|---|---|---|---|---|
| *Fílter et al. (2022)* | **Run up VJ** *(Control)* | NA | **3D motion-tracking opto-electronic camera system (8-camera)** | **Jump performance**: Jump height | **Run up VJ** showed ↑ horizontal velocity during initial contact, rate of force development, total impulse during push-off, COM horizontal and resultant velocity during take-off and VJ performance. |
| | **Run up VJ + ball:** | | Oqus motion analysis (Qualisys, Gothenburg, Sweden) | **Kinematic variables**: Pelvic torsion, COM velocity | **Run up VJ** showed ↓ CT. |
| | | | Frequency: 200 Hz; Markers: 64—type NA | **Kinetic variables**: RFD, impulse. | |
| | | | **Force platform (x1)** | Kinetic data were not normalized | |
| | | | Kistler (Kistler Instruments, Winterthur, Switzerland) | | |
| | | | Frequency: 200 Hz; Axes: triaxial | | |
| *Fischer et al. (2021)* | **DVJ** *(Control)* | Errors & time & Lead Times and Accuracy Scores (score = [correct number − incorrect number] time to complete) | **3D motion-tracking opto-electronic camera system (10-camera)** | **Kinematic variables**: Knee angles | **DVJ DM, DVJ Recall, DVJ Recall + Attention** and **DVJ Recall + Attention DM** showed ↓ peak knee flexion angle compared **DVJ**. |
| | **DVJ DM** | | (Motion Analysis, Santa Rosa, CA, USA) | **Kinetic variables**: Knee Abduction Moment | |
| | **DVJ Recall** | | Frequency: 250 Hz; Markers: 14 retro-reflective | Kinetic data were normalized by body mass and height. | |
| | **DVJ Recall + Atention** | | **Force platforms (x2)** | | |
| | **DVJ Recall + Attention DM** | | OPT464508-2K (Advanced Mechanical Technology, Watertown, MA, USA) | | |
| | | | Frequency: 1000 Hz; Axes: triaxial | | |

| Author, year | Testing conditions | Performance/ errors in secondary task | Assessment instruments (analysis systems) | Variables | Main findings |
|---|---|---|---|---|---|
| *Ford et al. (2017)* | **DVJ + Overhead goal** *(Control)* | NA | **3D motion-tracking opto-electronic camera system (14-camera)** | **Jump performance:** Jump height | **DVJ + Virtual Overhead goal** showed ↑ hip extensor moment and hip angular impulse. |
| | **DVJ + Virtual Overhead goal** | | Raptor-12 (Motion Analysis, Santa Rosa, CA, USA) | **Kinematic variables:** Trunk and lower limb joint angles | **DVJ + Virtual Overhead goal** showed ↓ trunk flexion. |
| | | | Frequency: NA; Markers: 43 retro-reflective | **Kinetic variables:** Peak lower limb moments during the ground contact. Kinetic data were not normalized | |
| | | | **Force platforms (x2)** | | |
| | | | AMTI BP600900 (Advanced Mechanical Technology, Watertown, MA, USA) | | |
| | | | Frequency: NA; Axes: triaxial | | |
| *Imai et al. (2022)* | **DVJ** *(Control)* | NA | **3D motion-tracking opto-electronic camera system (8-camera)** | **Kinematic variables:** Lower limb angles | **DVJ + calculation** showed ↑ peak knee abduction moment on both limbs, moments of hip and ankle joints and vGRF. |
| | **DVJ + calculation** | | Oqus motion analysis (Qualisys, Gothenburg, Sweden) | **Kinetic variables:** vGRF, Peak lower limb joint moments | |
| | | | Frequency: 120 Hz; Markers: 46 retro-reflective | Kinetic data were normalized by body mass | |
| | | | **Force platforms (x2)** | | |
| | | | Bertec AM6110 (Bertec, Columbus, OH, USA) | | |
| | | | Frequency: 600 Hz; Axes: triaxial | | |
| *Kajiwara et al. (2019)* | **SDL** *(Control).* | NA | **3D motion-tracking opto-electronic camera system (16-camera)** | **Kinematic variables:** Lower limb angles | **SDL Stroop** test showed ↑ maximum tibial internal rotation angle. |
| | **SDL Stroop test** | | MX-T20 Vicon Motion System (Vicon, Oxford, UK) | **Kinetic variables:** peak GRF | |
| | | | Frequency: 100 Hz; Markers: 23 retro-reflective | **Others:** Muscle activity of lower limb muscles | |
| | | | **Force platform (x1)** | Kinetic data were not normalized | |
| | | | Accugait (Advanced Mechanical Technology, Watertown, MA, USA) | | |
| | | | Frequency: NA; Axes: triaxial | | |
| | | | **EMG (x6 wireless electrodes)** | | |
| | | | Trigno Wireless EMG (Delsys, Boston, MA, USA) | | |

| Author, year | Testing conditions | Performance/ errors in secondary task | Assessment instruments (analysis systems) | Variables | Main findings |
|---|---|---|---|---|---|
| *Lin et al. (2020)* | **DVJ + Over-head goal** *(Control).* | NA | **3D motion-tracking opto-electronic camera system (10-camera)** | **Kinematic variables**: Lower limb angles | **DVJ + Overhead goal DM** showed ↓ knee flexion and ↑ in vGRF, loading rate and stability index. |
| | **DVJ + Over-head goal DM** | | MX-13+ Vicon Motion System (Vicon, Oxford, UK) | **Kinetic variables:** vGRF | |
| | | | Frequency: 200 Hz; Markers: NA | **Other:** Stability index, Loading rate | |
| | | | **Force platform (x2)** | Loading rate was calculated as normalized peak vGRF to body mass divided by time to peak vGRF. | |
| | | | Kistler 9260AA6 (Kistler Instruments, Winterthur, Switzerland) | | |
| | | | Frequency: 1000 Hz; Axes: tri-axial | | |
| *Ren et al. (2022)* | **DL** *(Control)* | NA | **3D motion-tracking opto-electronic camera system (10-camera)** | **Kinetic variables:** vGRF in three directions | **DL Visual Cognitive** showed ↑ TTS, Stability Index and COP. |
| | **DL visual-cognitive** | | Vicon Motion System (Vicon, Oxford, UK) | Kinetic data were normalized by body mass | |
| | | | Frequency: NA; Markers: NA, retro-reflective | **Others:** COP, Stability index, TTS | |
| | | | **Force platform (x2)** | | |
| | | | Kistler 9287B (Kistler Instruments, Winterthur, Switzerland) | | |
| | | | Frequency: NA; Axes: triaxial | | |
| *Richwalski et al. (2018)* | **DL side/Jump** *(Control)* | NA | **3D motion-tracking opto-electronic camera system (12-camera)** | **Kinetic variables:** vGRF, Lower limb moments, Power absorption | **DL side/Jump DM** showed ↑ vGRF |
| | **DL side/Jump DM** | | Vicon F40 Motion System (Vicon, Oxford, UK) | **Kinematic variables:** Lower limb angles | **DL side/Jump DM** showed ↓ knee and hip absorption and knee adduction moment. |
| | | | Frequency: 200; Markers: 40 retro-reflective | Kinetic data were normalized by body mass | |
| | | | **Force platforms (x2)** | **Others:** Loading rate | |
| | | | AMTI OR6-7-2000 (Advanced Mechanical Technology, Watertown, MA, USA) | | |
| | | | Frequency: 2000; Axes: triaxial | | |

| Author, year | Testing conditions | Performance/ errors in secondary task | Assessment instruments (analysis systems) | Variables | Main findings |
|---|---|---|---|---|---|
| *Stephenson et al. (2018)* | **DL side/Jump** *(Control)* | Standard deviation of the signal timing | **3D motion-tracking opto-electronic camera system (8-camera)** | **Jump performance:** Jump height, jump distance, RSI | **DL Side/Jump DM and DL Side/Jump DM300** showed differences in knee joint angles and moments. |
| | **DL side/Jump DM** | | Vicon Bonita Motion System (Vicon, Oxford, UK) | **Kinetic variables:** Lower limb moments | ↑Time to react (**DL side/Jump (Control)** > **DL side/Jump DM** > **DL side/Jump DM300** > **DL side/Jump DM150** > **DL side/Jump Dmlanding**): knee moments ↓ for medial jump directions and ↑ lateral jump direction. |
| | **DL side/Jump DM300** | | Frequency: 160; Markers: 23 retro-reflective | **Kinematic variables:** Lower limb angles | |
| | **DL side/Jump DM150** | | **Force platforms (x2)** | Kinetic data were normalized by body mass and height | |
| | **DL side/Jump DMlanding** | | Bertec 4060 (Bertec, Columbus, OH, USA) | **Others:** signal timming, ST | |
| | | | Frequency: 1600 Hz; Axes: triaxial | | |
| *Wilke et al. (2021)* | **CMJ number recall 1** | Landing errors & recall errors (*n*) | **Capacitive pressure platform** | **Kinetic variables:** pGRF | ↑Cognitive load (**CMJ number recall 3** > **CMJ number recall 2** > **CMJ number recall 1**): ↑ visual distraction (recall errors) and mediolateral COP. |
| | **CMJ number recall 2** | | Zebris FDM (Zebris Medical, Isny, Germany) | Kinetic data were not normalized | |
| | **CMJ number recall 3** | | Frequency: 50 Hz; Axes: One (vertical) | **Others:** COP, TTS, | |

**Notes.**

NA, Not available; EMG, Electromyography; MVC, Maximum voluntary contraction; RFD, Rate of force development; JH, jump height; iKVA, initial knee valgus angle; iKFA, initial knee flexion angle; iKIRA, initial knee internal rotation angle; pKVA, peak knee valgus angle; dKFA, knee flexion angular displacement; pKIRA, peak knee internal rotation angle; pKVM, peak knee varus moment (torque); pKEM, peak knee extension moment (torque); pKERM, peak knee external rotation moment (torque); ST, stance time; GRF, Ground reaction force; COP, center of pressure; COM, center of mass; TTS, Time to stabilization; ↑, increase; ↓, decrease.

displacement of the center of mass (*Akbari, Kuwano & Shimokochi, 2023*) and contact times with the platform (*Almonroeder, Garcia & Kurt, 2015*; *Fílter et al., 2022*; *Beardt et al., 2018*) decreased. This suggests that the inclusion of a constraint or coordinative difficulty tends to increase the stiffness of jump-landing. Unlike tests with cognitive DT (motor-cognitive), coordinative DT (motor-motor) were generally more specific to the sports modality; some studies even used virtual reality to simulate action in a real sports context (*DiCesare et al., 2020*; *Ford et al., 2017*). A large number of studies that have introduced a coordinative (motor-motor) DT have included a sport-specific mobile in the

task, for example heading (*Akbari, Kuwano & Shimokochi, 2023*; *Fílter et al., 2022*; *DiCesare et al., 2020*; *Alanazi et al., 2020*); grabbing (*Almonroeder et al., 2018*; *DiCesare et al., 2020*; *Ford et al., 2017*); or spiking a ball (*Beardt et al., 2018*).

Despite the heterogeneity regarding cognitive dual-tasking and assessment systems employed across studies, all included studies that utilized cognitive DT during jump and landing tasks (motor-cognitive) reported significant changes in some key biomechanical variables related to injury risk factors (*Kajiwara et al., 2019*; *Wilke et al., 2021*; *Imai et al., 2022*; *Fischer et al., 2021*; *Ren et al., 2022*; *Ford et al., 2017*). Specifically, an increase in peak vGRF during landing was observed (*Imai et al., 2022*), as well as increases in TTS and Stability Indexes in different axes (*Ren et al., 2022*), and an increase in the COP (*Wilke et al., 2021*; *Ren et al., 2022*). Additionally, an increase in hip extension (*Imai et al., 2022*; *Ford et al., 2017*), knee abduction (*Imai et al., 2022*), and ankle plantar flexion torques were observed (*Imai et al., 2022*). Kinematically, increases in peak tibial internal rotation angle were described (*Kajiwara et al., 2019*), along with a reduction in peak knee flexion angle (*Fischer et al., 2021*) and trunk flexion angle (*Ford et al., 2017*). Regarding the difficulty level of the cognitive task employed, it is important to note that only one of the included articles studied different levels of complexity in this task (*Wilke et al., 2021*). This could be considered a relevant factor to consider, as previous studies have suggested that postural control in easy cognitive tasks tends to increase due to a shift in attention focus, while more demanding tasks tend to impair postural control, showing an inverted "U" relationship between task difficulty and control (*Huxhold et al., 2006*). As for the type of cognitive tasks proposed in the included studies, they are generally nonspecific (see Table 4) with respect to the type of tasks, level of unpredictability, and cognitive demands that can be encountered in team sports practice (*Fuster, Caparrós & Capdevila, 2021*; *Gonçalves et al., 2016*). The most specific tasks within the included studies are those related to identification and memorization of opponents' numbers (*Wilke et al., 2021*), and visual tracking of spheres interacting in a three-dimensional space (*Ren et al., 2022*). In future research in this area, it would be interesting to incorporate tasks that more accurately reflect the cognitive demands inherent in team sports or based on the player's position and role (*Viñas et al., 2023*). Additionally, proposing variations in the difficulty levels of these tasks to adapt them to different contexts and needs would be beneficial.

Another factor to consider in the review is the effect that DM has on the biomechanical changes of the lower limb, when an athlete chooses between different motor responses based on external stimuli and the evaluation of information in a limited time. Of the studies included in this review, there are seven (*Lin et al., 2020*; *Kajiwara et al., 2019*; *Stephenson et al., 2018*; *Almonroeder et al., 2018*; *Fischer et al., 2021*; *Beardt et al., 2018*; *Richwalski et al., 2018*) that have specifically analyzed the effect of including DM (conditioned discriminative response with reaction time) in comparison to the same action without DM as a control. Overall, significant changes in various biomechanical variables of the lower limbs were observed in all studies due to the inclusion of DM. Specifically, several studies found lower peak angles of hip flexion (*Beardt et al., 2018*), and knee flexion (*Lin et al., 2020*; *Almonroeder et al., 2018*; *Fischer et al., 2021*; *Beardt et al., 2018*), along with increased knee abduction (*Almonroeder, Garcia & Kurt, 2015*) and tibial internal rotation (*Kajiwara et al.,*
2019). Additionally, increases in vGRF were also reported (*Almonroeder, Garcia & Kurt, 2015*; *Lin et al., 2020*). In contrast to these results, one of the included studies showed an increase in knee internal rotation torques and peak vGRF in the control condition compared to the DM condition when the task was pivoting on the dominant limb upon landing (*Richwalski et al., 2018*). The results of this study do not align with previous studies that have investigated the effects of anticipated *vs.* unanticipated actions (*Brown, Brughelli & Hume, 2014*; *Brown, Palmieri-Smith & McLean, 2009*; *Borotikar et al., 2008*; *Houck, Duncan & Kenneth, 2006*). The authors of the study suggested that the observed differences may be due to the movement following landing that required the participant to move 90° laterally, whereas other studies look at anterior translational movement (*e.g.,* running, long jump) that were controlled prior to landing (*Akbari, Kuwano & Shimokochi, 2023*; *Fílter et al., 2022*; *Richwalski et al., 2018*). In any case, it is shown that including a DM condition produced significant changes in joint mechanics, altering the preplanned movement pattern. Lastly, only one of the included studies evaluated the effect of modifying the anticipation time of the motor response to be performed (*Stephenson et al., 2018*). They found that the kinematic and kinetic parameters of the knee upon landing vary depending on whether the response is anticipated or not, as long as there is enough time for the reaction to be adequate and evoke the preplanned movement pattern (probable threshold at 300 ms). It is noteworthy that the type of stimulus and the possible motor responses in most of the DM studies included are non-specific to the sport (Table 4), with the exception of the study by *Beardt et al. (2018)*, in which the specific actions to be performed by the participants were conditioned by an opponent.

The systematic review conducted has allowed for a detailed exploration of different studies on the biomechanical performance of the lower extremities in contexts of DT or jump-landing tasks conditioned by added constraints. Focusing on performing a secondary task correctly potentially diverts attentional resources, adversely affecting the coordinative motor control necessary for optimal execution of the primary task (*McIsaac, Lamberg & Muratori, 2015*). This phenomenon resembles real-life situations in sports, where attention is divided among multiple simultaneous stimuli, such as making accurate passes, eluding opponents, or performing decoy movements; and where attention is focused precisely on performing the secondary task during jump-landing (passing, receiving, decision making) as correctly as possible, without errors. In this context, a recent review highlights measuring errors in the secondary task as a critical factor that could significantly influence performance and biomechanical execution of the primary task in DT conditioned tests (*Chaaban, Turner & Padua, 2023*). However, it is notable that of the fifteen studies included in this review, only four have specifically addressed trial-error measurement or performance in the secondary task, through recording the number of attempts (*Almonroeder et al., 2018*), errors, time, reaction times, and accuracy scores (*Fischer et al., 2021*; *Stephenson et al., 2018*), as well as landing errors and memory errors (*Wilke et al., 2021*). The absence of recording these errors in most of the reviewed studies may introduce bias in the results, as they focus exclusively on evaluating the primary task without considering the efficiency in managing simultaneous tasks. This omission not only deviates from real competition conditions but also may lead to biased conclusions regarding the effect of such tasks on

biomechanical variables and injury risk factors in team sports, emphasizing the importance of incorporating these parameters in future research.

Another notable aspect of the systematic review is the remarkable shortage of comparative analysis between lower limbs in the examined literature. Most studies included in this review gather biomechanical data solely from one limb in unilateral jumps performed with the subjectively dominant limb (*Lin et al., 2020*; *Kajiwara et al., 2019*; *Stephenson et al., 2018*; *Akbari, Kuwano & Shimokochi, 2023*; *Richwalski et al., 2018*) or non-dominant limb (*Almonroeder et al., 2018*), or in bilateral jumps (*Wilke et al., 2021*; *Fílter et al., 2022*; *DiCesare et al., 2020*; *Alanazi et al., 2020*; *Ren et al., 2022*; *Ford et al., 2017*). In two of the studies included in the review, the effects are measured biomechanically in both limbs, but no statistical comparisons are made between them (*Fischer et al., 2021*; *Beardt et al., 2018*). Only one study conducts the analysis of different variables between the dominant and non-dominant limb (*Imai et al., 2022*). The scarcity of direct comparisons significantly limits our understanding of biomechanical symmetry between limbs during task execution in DT conditions, crucial for identifying potential imbalances that may increase injury risk. This review underscores the need to incorporate comparative analyses between the dominant and non-dominant limb, as well as between previously injured and uninjured limbs, in future research to advance toward a more precise evaluation of biomechanical function in sports and clinical contexts.

The clinical implication derived from the observed biomechanical changes in the reviewed studies underscores a direct relationship between the performance of DT and the increased risk of lower extremity injuries in team sports. These studies focus their analysis on kinetic and kinematic alterations during DT in jump-landing tests, identifying variables associated with the risk of anterior cruciate ligament (ACL) injury as the primary focus of attention. It has been observed that team sports athletes exhibiting significant knee abduction torque during landing have a heightened risk of ACL injuries (*Hewett & Myer, 2011*). Evidence suggests that a decrease in trunk neuromuscular control contributes to an increase in knee abduction torque, thereby elevating the risk of ACL injury (*Hewett & Myer, 2011*). Furthermore, it is highlighted that a reduction in knee flexion leads to an increase in the angle between the patellar tendon and the tibial axis, generating greater anterior tibial loading and consequently a higher risk of ACL injury (*Norcross et al., 2013*). When knee flexion angle is insufficient, the ability of the flexor musculature to provide posterior force to the tibia is compromised, increasing shear force on the anterior tibia (*Shimokochi & Shultz, 2008*). In relation to the above, elevated vGRF, coupled with reduced knee and hip flexion, indicates a relatively rigid landing pattern, which may increase forces acting on the ACL. This could lead to joint instability, especially when neuromuscular control is deficient (*Lin et al., 2020*). Therefore, the collective results of the various studies included in the review suggest a biomechanical profile of the lower extremity with an increased risk of injury in team sports.

Based on the combined results of the studies included in the review, it is evident that conducting standard bilateral jump-landing tests under controlled and stable conditions to assess team sports athletes entails deficiencies in external validity to understand what occurs in situations of greater specificity or competition. Such assessment tests appear

not to reveal biomechanical patterns indicative of high injury risk, which may emerge when tests are performed under high cognitive, coordinative, or temporally conditioned demands and/or DM. Therefore, the evaluation tests used to determine the risk of injury or recurrence, or to support the return to play in team sports, should incorporate specific cognitive, coordinative, or DM elements that simulate the specific demands inherent in team sports.

Finally, this systematic review is not without limitations. Numerous publications exist regarding the execution of motor tasks with varying levels of complexity, either in isolation or in combination with other tasks. A comprehensive search strategy, detailed in the materials and methods section, was implemented. However, it is possible that some relevant studies addressing the aims of this review may exist, albeit without explicit reference to the specific terms employed in the search strategy. A notable challenge pertains to the diverse approaches and paradigms applied in the interpretation and examination of DT conditions. Notably, discrepancies arise in defining what constitutes a DT, with some definitions or authors categorizing certain scenarios as such, while others characterize them as singular tasks influenced by overlaid constraints. Consequently, not all studies incorporated into this review may align with every conceptual definition of DT. Another aspect to consider in this systematic review is that the majority of the research on this topic has been conducted with female athletes. We believe this might be due to the higher prevalence of non-contact ACL injuries in female team sports athletes compared to males (*Chia et al., 2022*), which may have driven research interest in this population. The overrepresentation of female participants in the included studies in this systematic review may constrain the extrapolation of the findings to male athletes.

## CONCLUSIONS

This systematic review contributes to the existing body of knowledge by specifically focusing on the effects of incorporating DT during jump-landing tests on biomechanical variables related to lower-limb injury risk in team sports. The inclusion of DT or constraints in jump-landing tests significantly alters biomechanical variables related to lower extremity injury risk in team sports. Generally, in the included studies, a stiffer landing at the joint level was observed when cognitive, coordinative, or DM tasks were included, resulting in increased vGRF and peak joint torques, a decrease in peak knee and hip flexion angles, and overall reduced stability and postural control. These biomechanical changes suggest a higher risk of injury when performing the task under such conditions.

Regarding the types of tasks used in the studies included in the present review, coordinative tasks generally involved incorporating equipment from the sports discipline into jump-landing or using simulated situations with virtual reality, thus having a certain specificity with respect to the sport. In contrast, most studies that introduced cognitive or DM aspects into the task used stimuli and constraints that were highly nonspecific to the sport. Additionally, the majority of studies focused on bilateral tests or the specific evaluation of one of the two limbs, without conducting comparative analyses between limbs (dominant *vs.* non-dominant or injured *vs.* uninjured). Lastly, few studies have

focused on recording errors in secondary tasks, which could bias the results obtained with the inclusion of the task or constraint.

## PRACTICAL APPLICATIONS

Looking ahead, research in the field of sports science should focus on implementing studies that more faithfully reflect the complex demands inherent in real sports practice in team sports. This implies, first and foremost, designing and utilizing tasks and constraints that specifically emulate sporting situations, with particular emphasis on tasks related to DM and cognitive processes. Additionally, it is imperative to analyze the effects of including DT and constraints on both limbs, conducting detailed comparisons between the dominant and non-dominant or injured limbs with respect to the uninjured limb. This approach would allow for a deeper understanding of motor and cognitive dynamics in sports, especially regarding injury prevention and recurrence reduction. Furthermore, it is crucial to record errors in secondary tasks to control potential biases and ensure the validity and applicability of the results obtained.

The findings of research on DT and constraints in jump-landing tasks have direct application in the professional practice of physical trainers and sports rehabilitators. The use of classic jump-landing tests, both in screening for injury risk factors and in assessment batteries for return to play, has shown significant limitations by not adequately simulating the specific demands of competition. These limitations result in an incomplete representation of the coordinative and motor patterns that, under real competition conditions, pose a higher risk of injury. Therefore, there is a need to develop and validate specific tests, adapted to each sport and situation, that faithfully reproduce the characteristics and demands of sports practice. The implementation of these tests, which should include different levels of difficulty, will facilitate a more precise evaluation of the athlete's functional status, promoting more effective and safer intervention and rehabilitation strategies aligned with the true demands and levels of unpredictability of team sports.

### Funding
The Research Group in Technology Applied to High Performance and Health, TecnoCampus, Universitat Pompeu Fabra, Mataró, Barcelona, Spain supported the APC of this article. No additional funding was received. The funders had no role in study design, data collection and analysis, decision to publish, or preparation of the manuscript.

### Grant Disclosures
The following grant information was disclosed by the authors:
The Research Group in Technology Applied to High Performance and Health, TecnoCampus, Universitat Pompeu Fabra, Mataró, Barcelona, Spain.

## Competing Interests

The authors declare there are no competing interests.

## Author Contributions

- Sara González-Millán conceived and designed the experiments, performed the experiments, analyzed the data, prepared figures and/or tables, authored or reviewed drafts of the article, and approved the final draft.
- Víctor Illera-Domínguez conceived and designed the experiments, performed the experiments, analyzed the data, prepared figures and/or tables, authored or reviewed drafts of the article, and approved the final draft.
- Víctor Toro-Román conceived and designed the experiments, analyzed the data, prepared figures and/or tables, authored or reviewed drafts of the article, and approved the final draft.
- Bruno Fernández-Valdés conceived and designed the experiments, authored or reviewed drafts of the article, and approved the final draft.
- Mónica Morral-Yepes performed the experiments, analyzed the data, authored or reviewed drafts of the article, and approved the final draft.
- Lluís Albesa-Albiol conceived and designed the experiments, authored or reviewed drafts of the article, and approved the final draft.
- Carla Pérez-Chirinos Buxadé conceived and designed the experiments, authored or reviewed drafts of the article, and approved the final draft.
- Toni Caparrós conceived and designed the experiments, authored or reviewed drafts of the article, and approved the final draft.

## Data Availability

This is a systematic review/meta-analysis.

## Supplemental Information

Supplemental information for this article can be found online at http://dx.doi.org/10.7717/peerj.17720#supplemental-information.

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
