# Peer review of "Effects of adding dual-task or sport-specific task constrains to jump-landing tests on biomechanical parameters related to injury risk factors in team sports: a systematic review"

_PeerJ, doi:10.7717/peerj.17720_

## Round 0.1 · original submission · Minor Revisions

Dear Author's

We would like to express our gratitude for submitting your manuscript entitled "Effects of adding dual-task or sport-specific task constrains to jump-landing tests on biomechanical parameters related to injury risk factors in team sports: a systematic review" to our journal.

The reviewers acknowledged the quality of your work and identified a few areas that could be enhanced to further strengthen your argument and contribution to the scientific community. Their suggestions and critiques were valuable, and thus, we kindly request that you revise your manuscript according to the reviewers' suggestions.

Should you have any questions or require further clarification regarding the revisions requested, please do not hesitate to contact us. We once again appreciate your work and collaboration in this process.

Best regards,

Alexandre Medeiros

Reviewer 1 ·

Basic reporting

No comment

Experimental design

Some points require clarification for better understanding.

Validity of the findings

No comment

Additional comments

Methodology:
- Kindly update the status in PROSPERO.
- In lines 148-149, please revise the initial eligibility criteria for enhanced comprehension.
- Regarding lines 161-169, please elucidate whether the review process of the eligibility criteria was conducted in a blinded manner and provide details on the methodology to ensure this.
- It is suggested that lines 178-195 are more suited for inclusion in the discussion section.
- What rationale underlies the decision to change the risk of bias assessment tool? In the PROSPERO registration, only reference to NHLBI was made.

·

Basic reporting

Since the authors have selected an interesting research topic, their work should be acknowledged. However, its novelty is a little bit compromised. Looking for the main databases, there are some review papers that already worked close on this thematic.
In general, the manuscript is well written. However, authors should reinforce the importance of this systematic review. The meaning is not clear. The authors should enhance the question that authors want to answer with this study. I expect that the specific comments can the manuscript (in attachment file).

Experimental design

Well conducted.

Validity of the findings

Methodological quality has some flaws. The authors should attempt to this:


L-196-199 – Quote a pair of references in sport sciences sytematic review that used NHLBI quality assessment.
Results
Table 3 – Information about the training volume should be included in this table.
Table 4 – Details about number of repetitions, intervals, should be included in each Jumping-Landing Test.
Table 5 – Details about the assessment instruments should be included. What kind of 3D cameras sytem? Optoelectronical or digital cameras? 3D or 3D force plate? In what kind of frequqnecy both sytems operated? Details about the variables should be included, por example COM velocity? x, y OR z plane? All kinetic variables were normalized by Body Weight?
L36-137 – What biomechanical variables? Authors should better describe this point.

Additional comments

General comments

Since the authors have selected an interesting research topic, their work should be acknowledged. However, its novelty is a little bit compromised. Looking for the main databases, there are some review papers that already worked closely on this thematic.
In general, the manuscript is well written. However, authors should reinforce the importance of this systematic review. The meaning is not clear. The authors should enhance the question that authors want to answer with this study. I expect that the specific comments can the manuscript.

Specific comments

Abstract – Include a sentence with the perspectives about the future studies conducting jumping landing tests.

Introduction

L88-89 – Describe those biomechanical variables from kinematics and kinetics areas.
L110-111 – The last sentence is redundant.
L122-129 – What its the meaning of this review? What authors wants to answer? The argument that “we have not found any reviews...” is not a justification to conduct a systematic review. The authors should insert the importance of this systematic review of a practical application, based on studies already published regarding DT/jumping tests and biomechanical variables.
What is the meaning of this research? What authors wants to answer? What´s the relevance of this study? This should be clear at the end of this section.

Material and Methods
Methodological quality
L-196-199 – Quote a pair of references in sport sciences systematic review that used NHLBI quality assessment.

Results

Table 3 – Information about the training volume should be included in this table.
Table 4 – Details about number of repetitions, intervals, should be included in each Jumping-Landing Test.
Table 5 – Details about the assessment instruments should be included. What kind of 3D camera system? Optoelectronic or digital cameras? 3D or 2D force plate? In what kind of frequency both sytems operated? Details about the variables should be included, for example COM velocity? x, y or z plane? All kinetic variables were normalized by body weight?
L36-137 – What biomechanical variables? Authors should better describe this point.

Discussion

L351-353- Very superficial this argument. Please, analyse all contexts involved in this review.
L363-382 – Why authors are discussing the methodological errors, once they were not presented in the results section?
The authors should include a para discussing about the kinetic, kinematic and eletromyography methods applied in each one of the studies.
L423-425- What is the novelty of this review?

Reviewer 3 ·

Basic reporting

. Abstract

. Line 29: I suggest the authors update the final search date.



Materials & methods

. Line 228: “secluded”



Discussion

. Line 359: 300 ms



References

. Check the name of the journals. Sometimes it appears abbreviated, sometimes in full version.

Experimental design

The manuscript is well written and very explicative, but some minor changes should be considered.

Validity of the findings

The manuscript can be relevant to the scientific and technical community. There is clear understanding of methods, results and discussion. The authors implemented rigorous methodological analysis, which implied clarity in results and reading discussion.

·

Basic reporting

No comment

Experimental design

No comment

Validity of the findings

No comment

Additional comments

Congratulations on your diligent work on this systematic review examining the effects of adding dual-task or sport-specific task constraints to jump-landing tests on biomechanical parameters related to injury risk factors in team sports. The manuscript is well-written and structured clearly. The methods are thoroughly explained in alignment with the Preferred Reporting Items for Systematic Reviews and Meta-Analyses (PRISMA) guidelines. The discussion is clear and provides deep insights. Before acceptance, I would recommend addressing the following few minor points for clarification.

Line 23: “The aim of this systematic review was to examine the influence of performing a dual task (DT) during jump-landing tests on biomechanical variables related to lower limb injury risk in team sports.” I suggest that the authors include not only the effects of dual-task (DT) during jump-landing tests but also the effects of sport-specific task constraints in the aim of the study, as these are included in the title of the paper.

Line 29 & 140: Could you elaborate on the decision to include only studies published after 2013? Clarifying if this choice was driven by particular trends or developments in the field would enhance the rationale behind your methodology.

Lines 33-35: Please consider revising the sentence on how articles were selected to improve clarity and grammatical accuracy. A suggested revision might be: "Of the 656 records identified, 13 met the established criteria. Additionally, 2 more articles were manually included after screening references from the included articles and previous related systematic reviews."

Lines 40-43: “The execution of decision making during the jump-landing action resulted in biomechanical changes such as lower peak angles of hip flexion and knee flexion, along with increased vertical ground reaction force, knee abduction, and tibial internal rotation were reported”. The placement of "were reported" in this sentence appears incorrect. For clarity, please consider removing it.

Line 40: Please introduce and consistently use the abbreviation "DM" for “decision-making” throughout the manuscript.

General Comment for the Abstract: Please specify the methods used to assess the risk of bias in the included studies, in accordance with PRISMA 2020 guidelines for abstracts.

Line 51: I suggest replacing "conditional" with "physical".

Line 110: “Moreover, prior studies suggest that when cognitive DT is performed during VJ, landing mechanics are affected”. Please add "a" before "cognitive DT"

Lines 143-146: Please consider removing the first and last parentheses from the search strategy for better clarity.

Lines 153-159: Could you clarify what "a different subject" (3) refers to and define "regular team sport practice" in the exclusion criteria (4)? These details will enhance the transparency and replicability of your study selection process.

Lines 164-165: “In the first step, the references were uploaded to Abstract, from where author screened independently the literature from titles, abstracts, and keywords”. The description of the first step in your review process is unclear. Please revise for clarity. For example, "In the first step, the authors independently screened the literature from titles, abstracts, and keywords after uploading the references to a management system."

Line 168: Could you clarify what is meant by "suitable articles"? Please define the criteria that made these articles suitable for further searches.

Lines 170-174: In the section detailing data extraction, the word "extracted" is used redundantly. To improve readability, consider modifying the sentence as follows: "… outcome measures (all results involving kinematic or kinetic variables) were systematically extracted and documented on a spreadsheet."

Lines 174-177: The phrase "independently validated" in the description of the data extraction process seems redundant, given the context of a rigorous validation procedure. Consider simplifying the sentence for clarity. For instance: "The data extraction process was conducted by one author (S.G-M) and subjected to a rigorous validation procedure, with double-checking by a second author (V.I-D) to ensure accuracy and consistency."

Line 204: Please ensure that reference 52 is formatted as a superscript in the text where it is cited, according to the journal’s citation style guidelines.

Line 242: The sentence contains repetitive use of "and." Please consider revising for clarity. For example: "A single study reported a low risk of bias in the blinding of participants, personnel, and outcome assessment."

Lines 253-254: It is noteworthy that within the 15 studies examined, nine exclusively involved female participants, whereas only one focused solely on male participants. Given this gender disparity, it would be beneficial to discuss in the "Discussion" section whether gender plays a role in the systematic review results. Additionally, please address why the majority of selected studies predominantly involve female participants. This analysis could offer insights into the applicability and generalizability of your findings.


Line 269: The term "encapsulates" in the description of Table 4 might be perceived as too formal or detailed. Consider using "summarizes" or "details" for a clearer and more straightforward expression. For example: "Table 4 summarizes the principal characteristics of the study protocols included in the review."

Line 293: The word "exert" seems to be extraneous and could be removed for smoother reading. Consider revising to: "The present article addresses a review of the effects of incorporating DT during the jumping or landing action, both coordinative (motor-motor), cognitive (motor-cognitive), or conditioned by decision-making, on the biomechanical variables linked to lower limb injury risk factors in team sports."

297: The term "team sport" should be pluralized to "team sports" to correctly reflect that multiple sports are being considered.
Lines 297-299: In the description of how the section is organized, the term "conditions" related to different types of DT is somewhat vague. Could you please specify what these "conditions" refer to? Providing a clearer definition will help readers understand the exact factors being discussed.

Line 384: Please ensure the consistent use of the term "limbs" throughout the manuscript, replacing any instances of "legs" where applicable.

Line 390: Please consider replacing "severely" with "significantly" in the text to modify the emphasis. The adjusted sentence could read: "The scarcity of direct comparisons significantly limits our understanding of biomechanical symmetry between limbs during task execution in DT conditions, crucial for identifying potential imbalances that may increase injury risk."

---

## Round 0.2 · accepted · Accept

Dear Dra. González-Millán

Thank you for your submission to PeerJ.

I am writing to inform you that your manuscript - Effects of adding dual-task or sport-specific task constrains to jump-landing tests on biomechanical parameters related to injury risk factors in team sports: a systematic review - has been Accepted for publication. Congratulations!

Reviewer 1 ·

Basic reporting

No comment

Experimental design

No comment

Validity of the findings

No comment

Additional comments

Thank you for answering my questions and making the required changes.